# Evaluation of the Photocatalytic Activity of Water-Based TiO₂ Nanoparticle Dispersions Applied on Historical Painting Surfaces



**Stefania Pasquale [1,*], Massimo Zimbone [2], Francesco Ruffino [1,2] , Giuseppe Stella [1] and Anna Maria Gueli [1]**

[1] PH3DRA Labs & CHNet-INFN Sec CT, Dipartimento di Fisica e Astronomia "Ettore Majorana", Università degli Studi di Catania, Via. S. Sofia 64, 95123 Catania, Italy; francesco.ruffino@ct.infn.it (F.R.); giuseppe.stella@dfa.unict.it (G.S.); anna.gueli@unict.it (A.M.G.)

[2] Consiglio Nazionale delle Ricerche, Istituto per la Microelettronica e Microsistemi (CNR-IMM), Via. S. Sofia 64, 95123 Catania, Italy; massimo.zimbone@ct.infn.it

\* Correspondence: stefania.pasquale@ct.infn.it

**Abstract:** This paper aims at assessing the use of nanomaterials in painting conservation and in cleaning practices that could be alternative to the traditional ones to overcome the limits of new green materials. Titanium dioxide nanoparticles have been spread and studied on historical painting surfaces with good results. In particular, the properties of TiO₂ nanoparticles have been considered useful for self-cleaning and protective purposes against the accumulation of dirt and dust that represents the first phase in deterioration of historical painting surfaces. TiO₂ nanoparticles, prepared in distilled water by Pulsed Laser Ablation in Liquids, were applied on painting mock-ups realized in the laboratory according to old recipes and using historical binders and pigments. The surface characteristics of the painting were investigated by contact angle measurement and by Atomic Force Microscopy. The optical and aesthetical compatibility of the colloidal dispersions with the painting was assessed by spectrophotometry, and then the cleaning efficiency was evaluated by discoloration of a dye under ultraviolet irradiation, at fixed time intervals, using colorimetric technique. Because of the high reactivity of nanoparticles, the possibility of degrading the painting surface, together with the chromatic marker, was examined by colorimetric measurements. The evaluation of the color changes is important for all the materials belonging to cultural heritage, especially painting, for which the color modification induced by protective and/or cleaning interventions could irremediably compromise the work of art.

**Keywords:** conservation; Pulsed Laser Ablation; Methylene Blue; self-cleaning

## 1. Introduction

Conservation science is one of the most complex topics in materials science, as it requires interdisciplinary expertise, especially for easel paintings, which are still today among the greatest challenges in conservation and restoration of cultural heritage. This is due to their composite and heterogeneous structure, constituted by both organic and inorganic materials and built up from canvas and/or wood, ground, paint, and varnish layers. Treating materials of this complexity requires skills acquired through extensive training and practical experience, especially in the cleaning intervention. Considering all methods, cleaning represents an irreversible treatment, and for this reason, it would be better to adopt certain measures that can prevent or slow down the accumulation of dirt and dust that represents the first phase in deterioration of the painting surface [1,2].

Recent developments have shown that the complex tasks of cultural heritage conservation can be performed very effectively using nanomaterials and nanotechnology methods [3–8]. For example, dispersions of calcium hydroxide nanoparticles have been used for restoration of lifted and flaking layers in wall paintings [9,10], and oxides and

hydroxides of alkaline earth metals have been employed for deacidification of wood, paper, and canvas [11].

In recent years, titanium oxide was also tested on different materials belonging to cultural heritage, with good results for self-cleaning, de-polluting, and self-sterilization of stone and mortar [12].

Concerning the use of titanium dioxide on paintings, debate is still open. It has been proposed that $TiO_2$ has both positive and negative influences on paint film durability. On one hand, adding $TiO_2$ to a specific paint may improve its durability due to $TiO_2$ protection and self-cleaning, but on the other hand, it may negatively impact the painting due to photocatalytic degradation of the organic components related to the binder. According to some studies, $TiO_2$ affects paint durability, especially if it is used as a pigment in paintings [13–16]. It was found, in fact, that when titanium oxide absorbs UV radiation, the generated electrons and holes have sufficient energy to induce the formation of reactive oxygen species. These extremely active oxidizing species are responsible for the decomposition of several organic compounds [17–19].

In this study, $TiO_2$ nanoparticles have been obtained by Pulsed Laser Ablation in Liquids (PLAL). This high-power laser pulse technique is focused on a solid titanium target immersed in water. Local temperature rise, induced by laser pulse, creates a plasma Ti plume that expands in the liquid. The subsequent Ti oxidize in water, forming $TiO_2$. The further cooling of the temperature due to the presence of water releases nanoparticles in the solution [20]. PLAL represents a versatile, cost-competitive, and "green" method to produce nanoparticles. It avoids the use of chemical reagents, and this is particularly important because, by simple water, it is possible to spread nanoparticles on peculiar surfaces such as paintings. For these types of polychrome materials, it is important to avoid altering the aesthetic and optical characteristics.

Furthermore, titanium dioxide nanoparticles produced by PLAL have properties, such as the presence of amorphous phase, that make them particularly useful [20]. For this reason, the photocatalytic activity of the Laser-Ablated nanoparticles has been compared with crystalline anatase nanoparticles.

The optical properties and the color compatibility of the $TiO_2$ dispersions have been investigated, by spectrophotometric technique, on the painting surfaces prepared in the laboratory using pigments representative of the most-used traditional pictorial technique [21,22]. Painting surfaces have been investigated by contact angle and roughness measurements by Atomic Force Microscope (AFM) before and after the $TiO_2$ nanoparticles spread. To evaluate the efficiency in terms of cleaning performance of the $TiO_2$, the mockups were dirtied with the Methylene Blue dye and then irradiated by UV at different times, according to the standard procedure [23]. Because of the impossibility of quantifying the photodegradation efficiency by the measurement of the dye concentration, in accordance with [23], the spectrophotometric technique has been used because it allows for quantifying color differences through the CIELCh color system, which is also useful in the evaluation of the color modifications on other materials belonging to cultural heritage [24–26].

Because of the sensitivity to UV irradiation of the pictorial materials [27], it is mandatory to monitor the possible color changes, according to European standard [28]. For this reason, a specific part of the research work is dedicated to assessing the drawbacks and potentialities of the use of nanoparticles on paintings for conservation protocols.

## 2. Materials and Methods

### 2.1. Colloidal Dispersion Preparation

The synthesis of Laser-Ablated (LA) nanoparticles was performed according to references [29–32]. Briefly, a Nd: YAG (Giant G790-30) 1064 nm laser (10 ns pulse duration, 10Hz repetition rate) was employed to irradiate titanium metal plate (Goodfellow, purity 99.99%). The laser was focused using a lens (focal length of 15 cm) on the bottom of a Teflon vessel filled with 5 mL of deionized Milli-Q water (resistivity 18 MΩcm). The sample was irradiated at a fluence of 5 J/cm$^2$, and the spot size was about 3.5 mm in diameter.

The mass of the ablated material was estimated by weighting the target before and after the ablation with a micro-analytical balance with a sensitivity of 100 µg. The titanium concentration in solution was calculated assuming that ablated material (1 mg) had been totally converted into nanoparticles in 5 mL of water. The prepared dispersed titania particles had a Feret diameter of 34 nm by Dynamic Light Scattering (homemade apparatus described in [29] and SEM (Gemini Zeiss SUPRA™ 25), and they were composed of a mixture of small crystallites and disordered $TiO_2$ [29]. The amorphous phase has important effects because it creates recombination centers and surface trap centers able to increase the probability of interfacial charge transfer. It also creates surface active sites for adsorption of molecules to be oxidized [29–32]. As demonstrated by [29–32], the nanoparticles obtained by PLAL exhibit photocatalytic activity comparable to that of commercial crystalline material. Commercial standard $TiO_2$ nanoparticles bought from Sigma-Aldrich (SA, St. Louis, MO, USA) were used as a reference for the discoloration experiments. The crystalline nanoparticles, composed by anatase phase, from Sigma Aldrich were characterized in a previous article [29]. The prepared solution was obtained by dissolving 1 mg of powder in 5 mL of Milli-Q water and stirring for 30 min.

In the following sections, we refer to Laser-Ablated $TiO_2$ nanoparticles as LA $TiO_2$ NPs and $TiO_2$ nanoparticles by Sigma-Aldrich as SA $TiO_2$ NPs.

### 2.2. Painting Mock-Ups Preparation

The "Bianco di San Giovanni" pigment was selected to prepare the painting mock-ups. This pigment is a lime-white pigment having mineral origin from calcium carbonate deposits [21,33]. The pigment, supplied in a state of dry bulk solid composed of loose particles with diameter range of $45 < \Phi < 75$ µm, was mixed in the ratio 1:3 with wet binders, according to old recipes [21,22]. The selected historical binders were casein, egg yolk, and linseed oil.

All the painting mock-ups were obtained by brushing the mixtures on canvas with a preparation layer constituted by a mixture of gypsum and rabbit glue prepared by Zecchi [33]. The pictorial layer completely obscured the canvas to obtain opaque paintings with a mass tone as defined by Mayer [22]. No varnishes were added. In this study, in fact, the indoor environment in which un-varnished paintings are exposed to light with UV emission was simulated. The dimensions of the mock-ups were 10 mm × 10 mm × 1 mm, including the canvas substrate.

### 2.3. Photocatalytic Tests

The colloidal dispersions were spread on the painting surface by syringe to give the same quantity to wet the painting. The photocatalytic efficiency was evaluated using a self-cleaning test using Methylene Blue (MB) dye [23]. The paintings were stained by a MB water solution (0.05 wt.%) using a syringe under dark condition, and then, after a 24 h long drying phase, the painting mock-ups were exposed to UV-A irradiation (368 nm) in controlled environment in Color Assessment Cabinet of VeriVide with irradiance value of $1.1 \pm 0.1$ mW/cm$^2$. The irradiation time was set by different steps of 17 h. This step was chosen after preliminary measurements aimed at verifying significant color changes.

### 2.4. Contact Angle and Roughness Measurements

To assess the water sensitivity of historical painting, different types of methods have been used. Contact angle measurement was used to define the surface wettability [34]. Water droplets were placed onto the material surface using a microsyringe, and the measurements were carried out to extrapolate the average value of contact angle. The roughness measurements have been performed by AFM.

The AFM analysis was performed by using a Bruker Innova microscope (Billerica, MA, USA) operating in high-amplitude mode. Ultra-sharpened Si tips were used (MSNL-10 from Bruker Instruments, Billerica, MA, United States, with anisotropic geometry, radius of curvature ~2 nm, tip height ~2.5 µm, front angle ~15°, back angle ~25°, side angle 22.5°,

nominal spring constant of 0.07 N/m). The AFM images were analyzed by using the SPMLABANALYSES V7.00 software to extract values for the RMS (Root Mean Square, i.e., standard deviation of the measured heights on 512 × 512 surface points).

*2.5. Optical and Colorimetric Characterization*

The spectrophotometric analysis on the painting mock-ups was conducted using a Konica Minolta® spectrophotometer, model CM-2600d with measurement geometry d/8°, selecting an area of 6 mm in diameter (SAV, Small Average Value). The results are related to the D65 illuminant and the CIE 1964 standard colorimetric observer (10° standard observer). The scale adjustment was performed using the white calibration plate (CM-A145) as a target for the maximum lightness and the device CM-A32 for the minimum lightness [35]. The data elaboration regarded SPEX/100 values (SPecular component EXcluded and UV included).

For evaluating the color changes before and after the surface treatment with titanium dioxide, the Spectral Reflectance Factor (SRF%) trend and the first derivative in function of the visible range (400–700 nm) were studied. To monitor any color modification of the painting, the CIELCh color space was considered [36,37]. This space allows evaluating the discoloration of the MB dye on the painting surfaces induced by titanium dioxide nanoparticles and the UV irradiation action. In particular, the chroma (C*) and the hue angle (h) were used, and they were calculated from the CIELAB color coordinates, as shown in the following Equations (1) and (2):

$$C^* = (a^{*2} + b^{*2})^{1/2} \tag{1}$$

$$H = \arctan (b^*/a^*) \tag{2}$$

where a* and b* are the color coordinates that quantify the red-green and yellow-blue hues, according to [24].

The hue angle has a value of 0° (or 360°) along the positive a* axis, a value of 90° along the b* axis, 180° along the negative a*, and 270° along the negative b* [37]. The Δh* value allows quantifying information on a* (red-green colors) and b* (yellow-blue colors) at the same time, and it represents the best quantity to evaluate the change in hue induced by the MB addition on the surface and after the photodegradation action due to $TiO_2$ NPs.

## 3. Results and Discussion

*3.1. Painting Surface Characterization after Nanoparticles Application*

The contact angle measurements were performed on a rough and chemically heterogeneous surface [38–40]. In Figure 1, the images acquired during the contact angle measurements are shown before (left) and after (right) the LA $TiO_2$ application for casein (a), egg tempera (b), and linseed oil (c) mock-ups.

The average values of contact angles were 62° ± 8° for casein painting, 124° ± 9° for egg tempera, and 130° ± 9° for linseed oil, before the titanium dioxide application. After, the contact angles changed to values lower than 90° for both tempera and linseed oil, showing a hydrophilic behavior of the treated surface with titanium dioxide nanoparticles.

The hydrophilicity agrees with the literature data about the use of titanium dioxide nanoparticles on other materials belonging to cultural heritage [41]. This should not be a problem in the indoor environment, and, for this reason, the application of titanium dioxide nanoparticles should be suitable for painting surfaces.

The AFM helped in the interpretation of the contact angle data. In Figure 2a,b, the images acquired during AFM scansions for casein and linseed oil mock-ups, before the nanoparticles application, are shown. The egg tempera painting surface was very inhomogeneous, and it was not possible to make the acquisitions with the AFM tip because of the surface conditions. The roughness evaluated in Root Means Square (RMS) was 500.9 nm and 705.7 nm for casein (Figure 2a) and linseed oil (Figure 2b), respectively. The

investigated area was 25 μm$^2$. The linseed oil surface presented a rough and flat area, probably due to the mock-up preparation, as visible in Figure 2b.

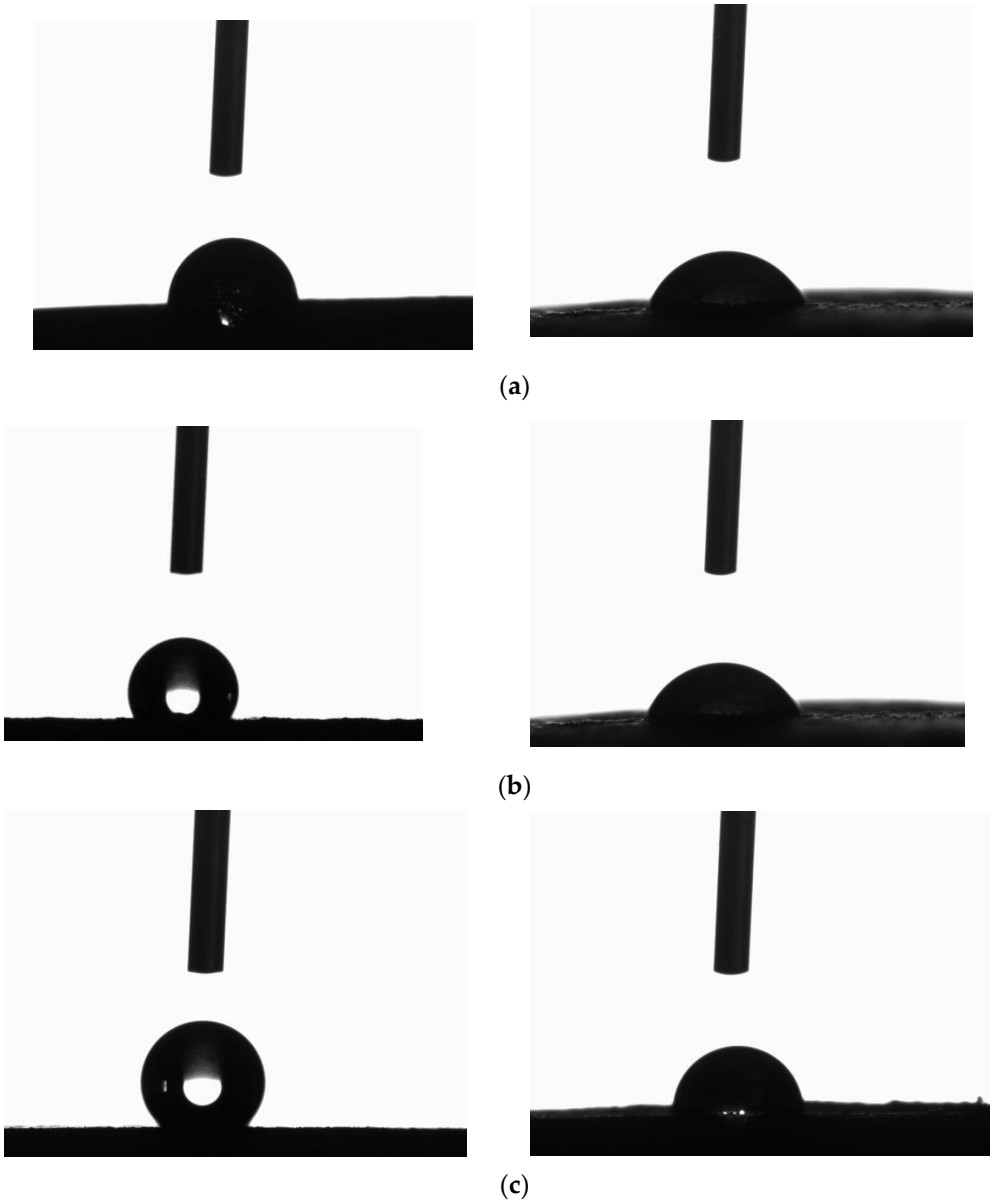

(a)

(b)

(c)

**Figure 1.** Images acquired during the contact angle measurement on the paintings prepared with the casein (**a**), the egg tempera (**b**), and the linseed oil (**c**), before (left) and after (right) the LA TiO$_2$ application.

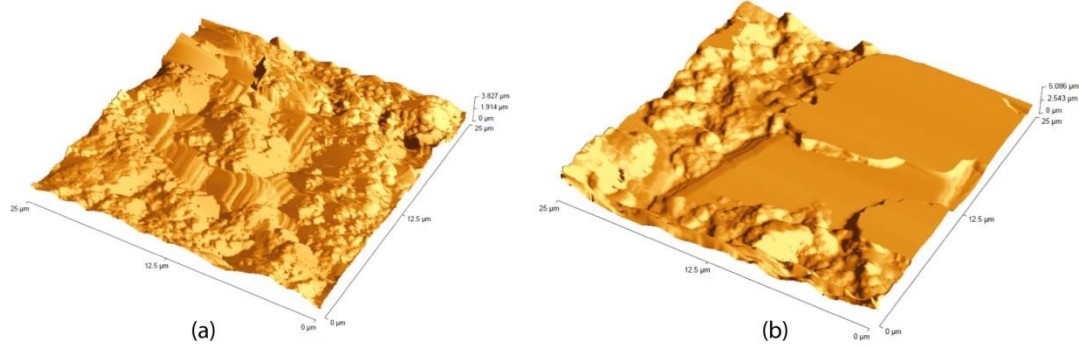

(a)

(b)

**Figure 2.** The AFM images acquired on casein (**a**) and linseed oil (**b**) paintings in an area of 25 μm$^2$.

In surfaces with these characteristics of roughness, the individuation of the nanoparticles is challenging. In Figure 3a,b, in an area 1 μm², the SA and LA nanoparticles are shown on linseed oil mock-ups.

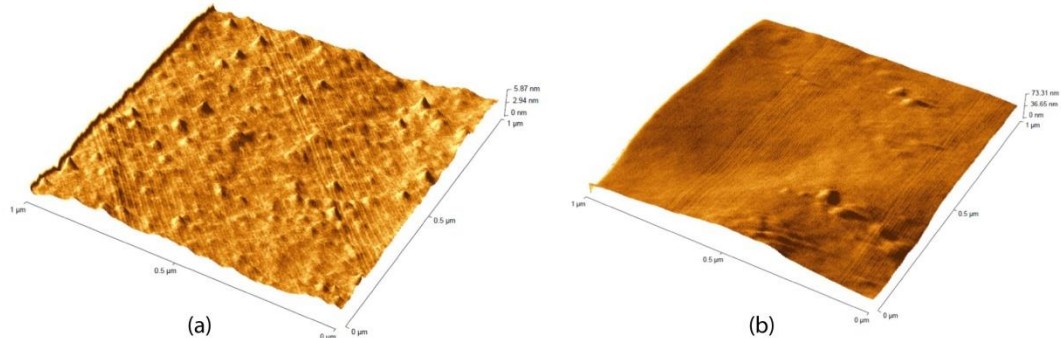

**Figure 3.** The AFM images acquired on 1 μm² linseed oil surfaces with SA (**a**) and LA (**b**) nanoparticles.

In Figure 4a–c, the SRF% trends and the related first derivative d(SRF%) vs. wavelengths (400–700 nm) for painting mock-ups prepared with casein, egg yolk, and linseed oil after and before the spreading of LA and SA TiO$_2$ NPs are shown. For all the types of mock-ups, considering the uncertainties of about 3%, no significant optical changes occurred after the application of LA and SA TiO$_2$ colloidal dispersion, as it is possible to see when also considering the first derivative of the spectral reflectance factor.

### 3.2. Photocatalytic Activity

The discoloration of the Methylene Blue, due to LA and SA TiO$_2$ NPs on painting surfaces, was confirmed by the study of the SRF% trends and the analysis of the color coordinates at different UV irradiation times.

In Figure 5, as an example, the SRF% trends in the visible region (400–700 nm) for the white painting prepared with linseed oil with LA and SA TiO$_2$ NPs are shown before and after the MB staining of the surfaces and during the UV irradiation (up to 106 h). After the MB application, the SRF% showed a greater reflection in the green region (500–600 nm) as a result of the mixing of the yellow surface (due to white pigments and the yellow linseed oil) and the blue of the dye. For linseed oil, the discoloration happened after 17 h of UV irradiation and gradually up to 102 h. The SRF% trends for both LA and SA TiO$_2$ NPs treated surface are homothetic.

The study of the SRF% trend allows acquiring optical information on the surface and must be correlated to the color changes quantification by the hue angle (h) of the CIELCh color space.

In the Figures 6–8, the h values are plotted against the UV irradiation time for casein, egg tempera, and linseed oil paintings. The h values are compared with those of the mock-ups stained with MB and treated with TiO$_2$ NPs. It is possible to observe that the h values for paintings with MB and with nanoparticles, increasing the UV irradiation time, become closer to the value of the reference painting, having the original color but with some differences. In the case of presence of nanoparticles, the MB degradation rate is also always enhanced in the case of casein, for which the painting with only MB has, at the end of the test (102 h), a Δh of about 30. The addition of TiO$_2$ induced a more incisive hue variation equal to 60.

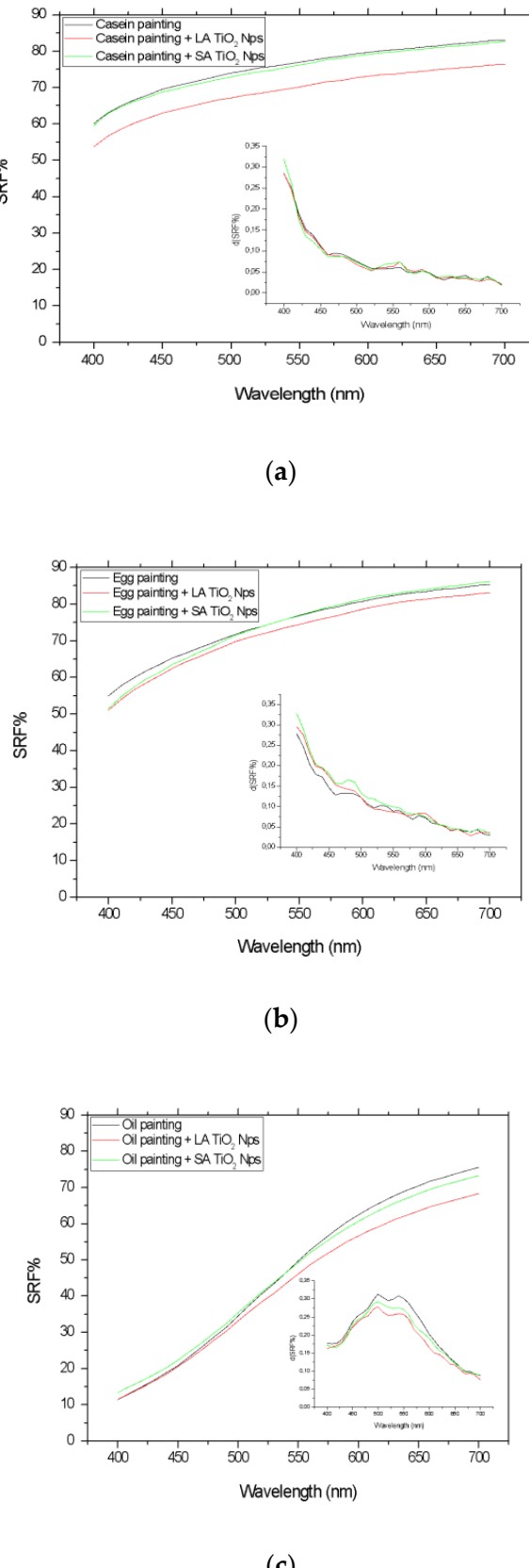

**Figure 4.** The SRF% and the related first derivative (d(SRF%) vs. visible region (400–700 nm) trends for casein (**a**), egg tempera (**b**), and linseed oil (**c**) painting after and before the LA and SA TiO$_2$ NPs deposition. The uncertainties are of about 3%, and they are not added in the graphs for a better legibility of the trends.

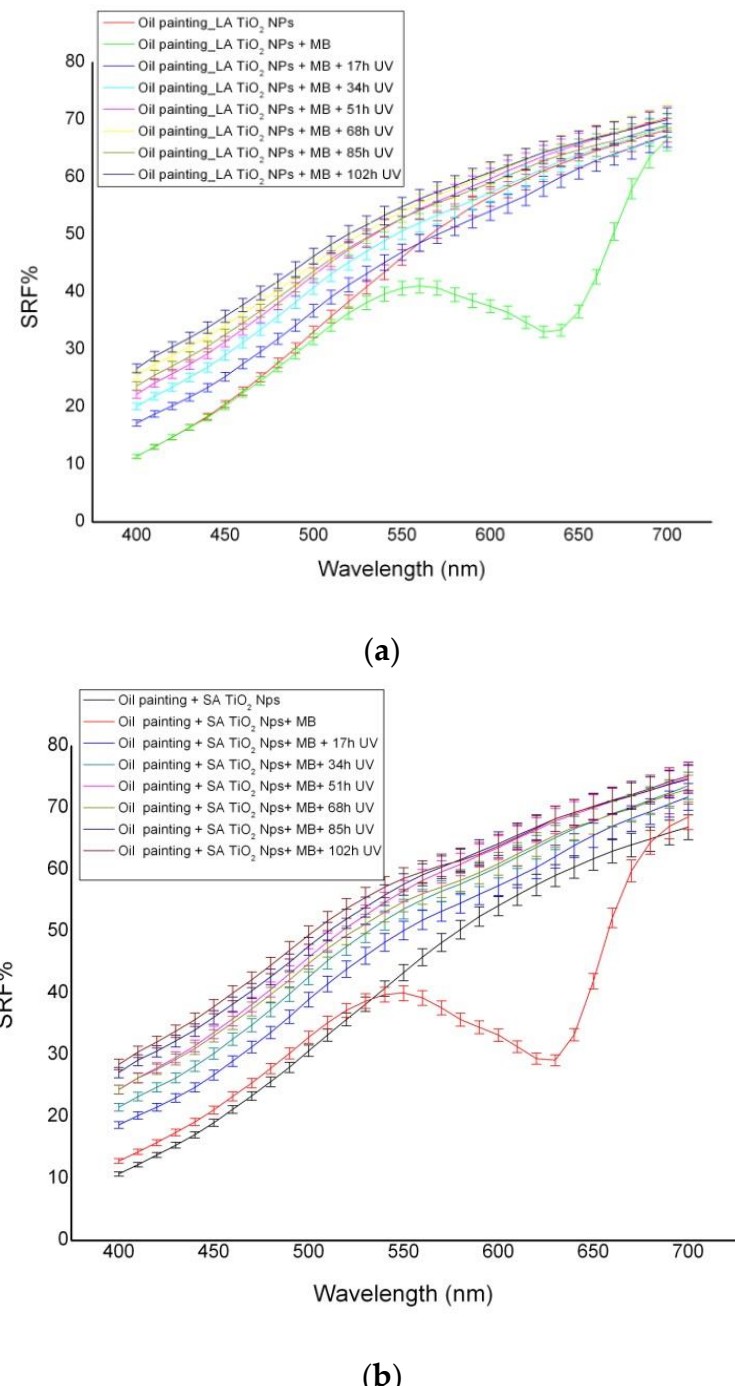

**Figure 5.** The SRF% versus the visible range (400–700 nm) trends for the painting prepared with linseed oil with LA (**a**) and SA (**b**) TiO$_2$ NPs at different UV irradiation times.

To compare the LA and SA TiO$_2$ NPs' self-cleaning activity on painting mock-ups, in the Figures 6b, 7b and 8b, the Δh values are calculated considering the different steps of UV irradiation with respect to the time 0. In general, the activities of the two types of nanoparticles are comparable, considering the uncertainties, during all the UV irradiation time. The results agree with what demonstrated by previous research articles in which the LA NPs presented photocatalytic activity comparable to that obtained with a highly active crystalline powder of commercially available TiO$_2$ [29–31].

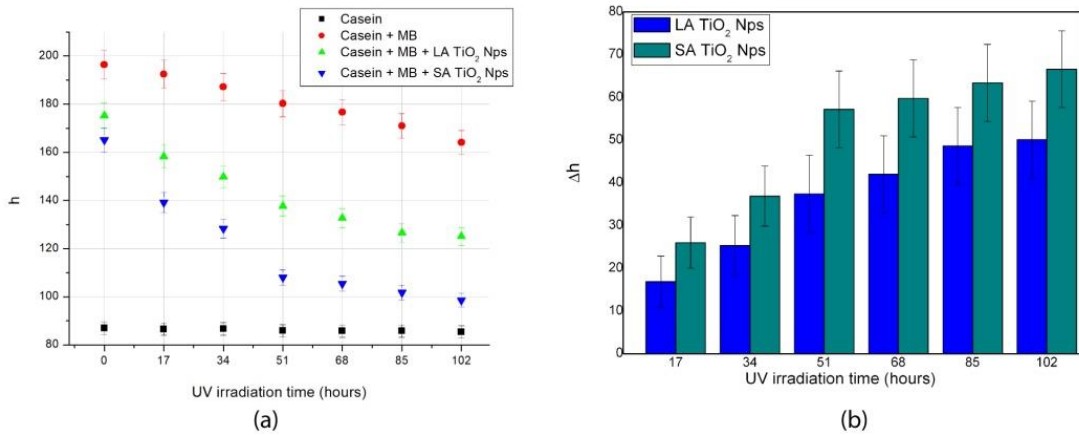

**Figure 6.** The h values for the casein painting stained with MB with and without LA and SA TiO$_2$ NPs (**a**) and the values of Δh calculated at the UV irradiation steps (**b**). The values reported the uncertainties calculated according to the propagation error theory.

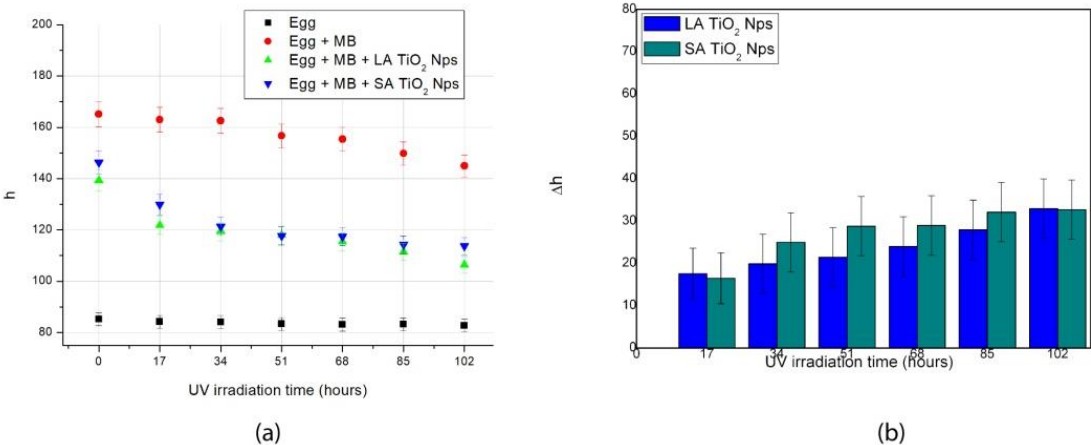

**Figure 7.** The h values for the egg painting stained with MB with and without LA and SA TiO$_2$ NPs (**a**) and the values of Δh calculated at the UV irradiation steps (**b**). The values reported the uncertainties calculated according to the propagation error theory.

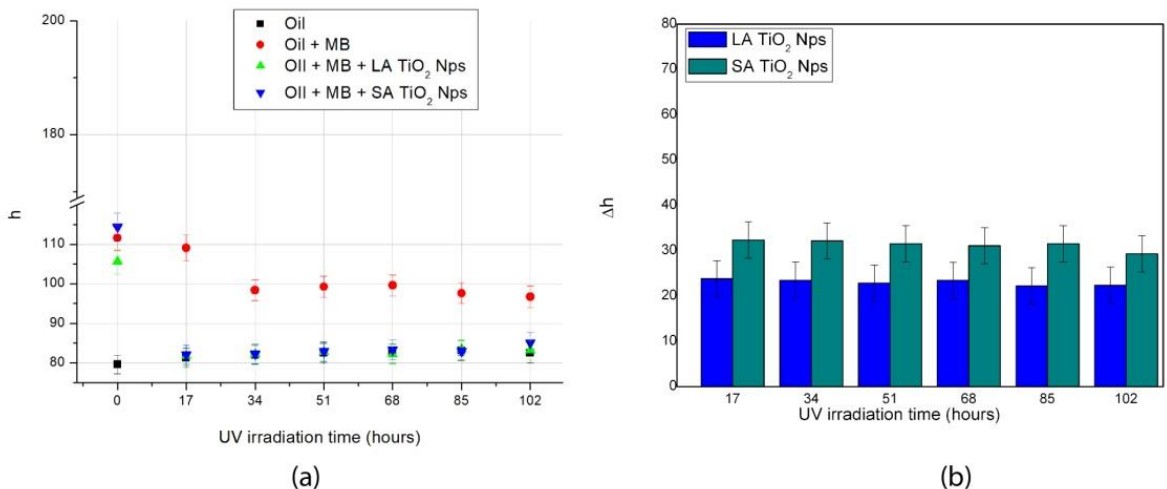

**Figure 8.** The h values for the oil painting stained with MB with and without LA and SA TiO$_2$ NPs (**a**) and the values of Δh calculated at the UV irradiation steps (**b**). The values reported the uncertainties calculated according to the propagation error theory.

The figures also highlight that some differences exist among the paintings. The best performance of MB degradation is observed for oil mock-ups (Figure 8). In this case, in fact, after 17 h the dye was already discolored, whereas for casein and egg paintings, the discoloration was slower (Figures 6 and 7). This could be explained considering that this binder is very reactive to UV irradiation [42] but, in general, this evidence must be further investigated, because it seems that the binder plays a role in the photocatalytic test. It could be due also to the differences in nanoparticle distribution on the different types of surface paintings.

To investigate the effect of the $TiO_2$ NPs over time, primarily to consider any possibility of degradation of the pictorial materials, in the following paragraph, the colorimetric changes induced by the UV irradiation on the painting mock-ups with $TiO_2$ NPs and without any dye to degrade are studied.

### 3.3. Painting Surface Degradation

The paintings are constituted by photosensitive materials, and, in an indoor environment, UV irradiation is not recommended by the UNI normative [27]. Nevertheless, many light sources include ultraviolet emission [37] that can be exploited to activate the photocatalytic activity of titanium dioxide. To monitor the color changes induced by ultraviolet irradiation during the photocatalytic test and to study the $TiO_2$ NPs action on the painting surfaces without any dye, the C* coordinate was plotted in the considered interval time (up to 102 h). The C* values, obtained by the color measurements, on the painting untreated and treated with LA and SA $TiO_2$ NPs, were normalized, and the results are plotted in Figure 9. The objective is to compare the color fading on paintings due to the action of the two types of titanium dioxide nanoparticles and due only to the UV action on the untreated painting. As visible in Figure 9, for all the paintings, a decrease in the saturation of the color, measured by C*, was registered. An exponential trend was used to fit the color coordinate with the function shown in Equation (3). C* is the color coordinate and a, b, and c are the scaling factor, the parameter related to exponential decay, and the offset, respectively.

$$C^* = a^* \exp^{(-x/b)} + c \tag{3}$$

Equation (3) is an empirical model, but it can find justification in literature. It is known, in fact, that the fading of dye or pigment often conforms to exponential decay behavior [43]. The values of the fit parameters, together with the R2 index, are shown in Table 1.

**Table 1.** The exponential fit parameters of the C* color coordinates, obtained by color measurements, are listed.

| Painting | a | b | c | $R^2$ |
|---|---|---|---|---|
| Casein | $0.50 \pm 0.02$ | $116.46 \pm 68.48$ | $0.95 \pm 0.02$ | 0.969 |
| Casein + LA $TiO_2$ | $0.52 \pm 0.02$ | $40.38 \pm 5.52$ | $0.49 \pm 0.02$ | 0.987 |
| Casein + SA $TiO_2$ | $0.71 \pm 0.05$ | $75.64 \pm 11.56$ | $0.31 \pm 0.05$ | 0.994 |
| Egg Tempera | $0.20 \pm 0.04$ | $144.59 \pm 44.24$ | $0.81 \pm 0.04$ | 0.994 |
| Egg Tempera + LA $TiO_2$ | $0.23 \pm 0.02$ | $32.32 \pm 7.49$ | $0.77 \pm 0.02$ | 0.973 |
| Egg Tempera + SA $TiO_2$ | - | - | - | 0.953 |
| Linseed Oil | $0.37 \pm 0.02$ | $45.86 \pm 5.34$ | $0.62 \pm 0.02$ | 0.992 |
| Linseed Oil + LA $TiO_2$ | $0.53 \pm 0.09$ | $161.32 \pm 39.54$ | $0.47 \pm 0.09$ | 0.997 |
| Linseed Oil + SA $TiO_2$ | $0.53 \pm 0.03$ | $67.51 \pm 10.11$ | $0.48 \pm 0.04$ | 0.993 |

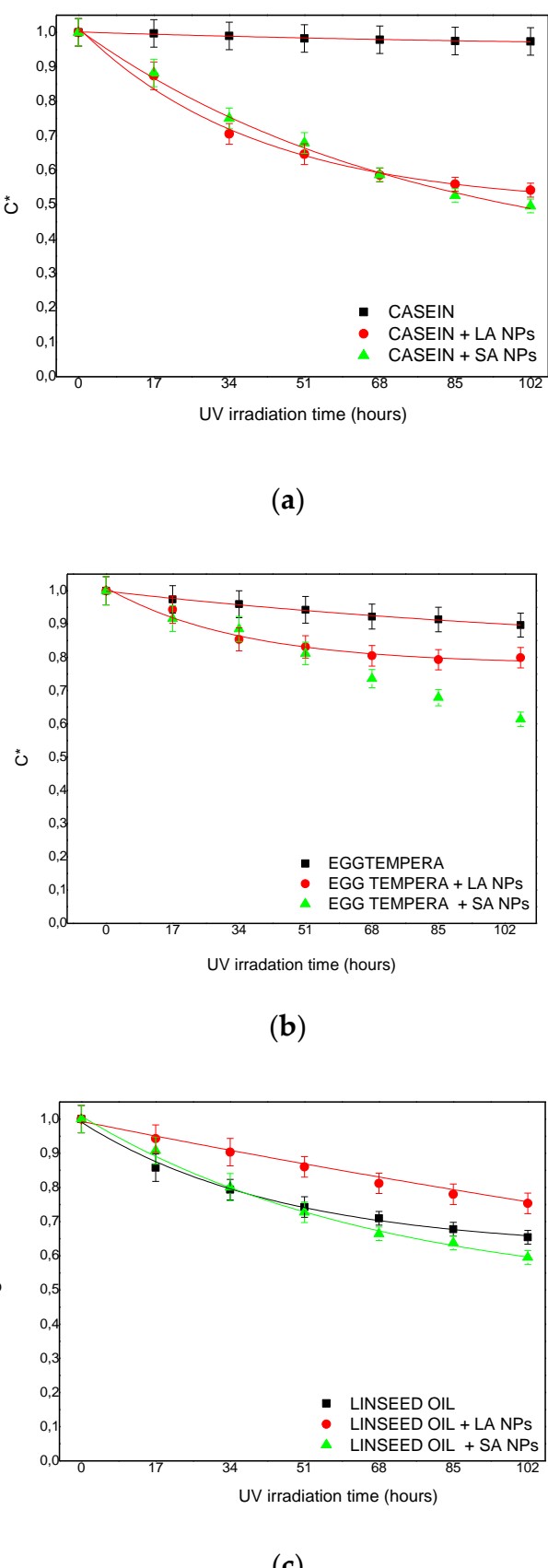

(**a**)

(**b**)

(**c**)

**Figure 9.** The trends of the C* coordinates, in the UV irradiation time, for the casein (**a**), egg tempera (**b**), and linseed oil (**c**) mock-ups untreated and treated with LA and SA TiO$_2$ NPs. The solid red lines are fitted curves to the experimental data over the whole time interval.

The decay b parameters listed in Table 1 were useful for interpreting the behavior of the paintings in terms of color fading. In the case of casein mock-ups, the LA $TiO_2$ NPs demonstrated a faster color fading than SA NPs, while in the case of linseed oil, the behavior was the reverse, being quicker for SA $TiO_2$ NPs. The egg tempera paintings presented a particular case, because the painting surface treated with SA $TiO_2$ NPs exhibited a trend that did not reproduce an exponential behavior. This could be due to the composition of this binder, which is constituted by whole egg and Arabic gum, according to the company [33]. This could have led to a complex degradation rate that also takes into account the contribution of more components. It seems that the empirical model of Equation (3) cannot fit the behavior of a more composite system than casein and linseed oil. The differences registered between the binders must be further studied. For these reasons, other investigations based on a long-term evaluation of the effect of the photocatalytic activity of $TiO_2$ NPs are ongoing by artificial ageing in controlled environments.

## 4. Conclusions

Titanium dioxide nanoparticles were applied on painting surfaces with protective and self-cleaning purposes. The nanoparticles, produced by PLAL, have a diameter of 30 nm and are composed of a mixture of small crystallites and disordered $TiO_2$.

After the application of nanoparticles using water as medium, the paintings were characterized by several methods. The surface of the mock-ups was investigated in terms of wettability with contact angle measurements after and before the nanoparticle application, and the expected hydrophilicity was observed. The surface roughness evaluated in RMS was 500.9 nm and 705.7 nm for casein and linseed oil, respectively, within the investigated area of 25 $\mu m^2$.

The study of the SRF% trends and the first derivative showed a good optical and aesthetical compatibility of the $TiO_2$ nanoparticles.

The self-cleaning efficiency of the Laser-Ablated nanoparticles applied on the painting mock-ups was compared with commercial ones, and high photocatalytic activity comparable to that obtained with a highly active crystalline powder of commercially available $TiO_2$ was observed. The comparison underlined that, in some cases, such as for linseed oil mock-up, after the first 17 h of UV irradiation, the Methylene Blue was completely degraded, as indicated by the h values. This suggests that the binder plays a role in the photocatalytic process that must be better investigated.

To understand if the painting, without the dye, could be subjected to degradation in terms of color fading, the C* coordinates trends in function of the UV irradiation time for painting with SA and LA $TiO_2$ NPs were investigated. In the time interval considered, the application of nanoparticles on a painting surface seems to not significantly affect the degradation of the painting in terms of color fading. Nevertheless, some differences have been registered due to the binders. For these reasons, other investigations are ongoing based on artificial ageing in controlled environments.

**Author Contributions:** Conceptualization, S.P. and A.M.G.; methodology, S.P., M.Z., F.R.; formal analysis, S.P.; data curation, S.P. and G.S.; writing—original draft preparation, S.P.; writing—review and editing, F.R. and M.Z.; supervision, A.M.G. and G.S. All authors have read and agreed to the published version of the manuscript.

**Funding:** This research was funded by Italian MIUR, grant number PON03PE_00214_2.

**Data Availability Statement:** The data presented in this study are available on request from the corresponding author.

**Conflicts of Interest:** The authors declare no conflict of interest.

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
