# Peer review of "Evaluation of the Photocatalytic Activity of Water-Based TiO2 Nanoparticle Dispersions Applied on Historical Painting Surfaces"

_heritage, doi:10.3390/heritage4030104_

Round 1

Reviewer 1 Report

The authors assessed the use of nanomaterials in painting conservation and in cleaning practices that could be alternative to the traditional ones to overcome the actual limits with new and green materials. They obtained TiO2 nanoparticles by Pulsed Laser Ablation in Liquids (PLAL) as well as investigated the color compatibility of the TiO2 dispersions on the painting surfaces and the efficiency in terms of cleaning performance of the TiO2. But the manuscript was prepared very carelessly, I don’t think the current version can be published in heritage. It is better to modify the manuscript according to the following suggestions:

  1. Some pages have too much blank, so the typesetting of the full text needs to be modified. And the pictures and annotations should preferably be on one page.
  2. It is unreasonable about the segmentation in the introduction section, please reconsider how to segment and revise the introduction part.
  3. There are many issues in the pictures. To be specific, there are several ways to write the serial numbers such as “a”, “a)” or “(a)”, and it’s better to adjust the position of the numbers to the upper left corner of the pictures. In addition, the details in the pictures need to be carefully revised and unified. Figures 5, 7 and 8 are fuzzy, please replace the image sources.
  4. Some pictures such as Figures 2 and 3 as well as Figures 6, 7 and 8 can be made to the group pictures, and the corresponding description sections don’t need to be divided into several paragraphs. Please reconsider the production of the group pictures in the article and carefully modify the description part.
  5. If possible, please provide the results of material and morphology characterization of the obtained TiO2
  6. In Figures 6-8, how to judge the self-cleaning activity of two kinds of TiO2 nanoparticles by the Δh values? And how do you know that the adhesive has an effect on the self-cleaning performance?
  7. There is no logic in the conclusion, and it’s difficult to see the key points that you want to express and a systematic summary of this research. Please clarify what this research ultimately intends to express, and systematically summarize the characterization results that can explain it.

Author Response

The authors assessed the use of nanomaterials in painting conservation and in cleaning practices that could be alternative to the traditional ones to overcome the actual limits with new and green materials. They obtained TiO2 nanoparticles by Pulsed Laser Ablation in Liquids (PLAL) as well as investigated the color compatibility of the TiO2 dispersions on the painting surfaces and the efficiency in terms of cleaning performance of the TiO2. But the manuscript was prepared very carelessly, I don’t think the current version can be published in heritage. It is better to modify the manuscript according to the following suggestions.

We thank the reviewer for the comments. They were useful to improve the paper. We replied below, point by point, to the questions.

  1. Some pages have too much blank, so the typesetting of the full text needs to be modified. And the pictures and annotations should preferably be on one page. -> We modified the layout.
  2. It is unreasonable about the segmentation in the introduction section, please reconsider how to segment and revise the introduction part. -> We revised the introduction.
  3. There are many issues in the pictures. To be specific, there are several ways to write the serial numbers such as “a”, “a)” or “(a)”, and it’s better to adjust the position of the numbers to the upper left corner of the pictures. In addition, the details in the pictures need to be carefully revised and unified. Figures 5, 7 and 8 are fuzzy, please replace the image sources -> We revised the figures.
  4. Some pictures such as Figures 2 and 3 as well as Figures 6, 7 and 8 can be made to the group pictures, and the corresponding description sections don’t need to be divided into several paragraphs. Please reconsider the production of the group pictures in the article and carefully modify the description part.-> We revised the Figures and the description parts.
  5. If possible, please provide the results of material and morphology characterization of the obtained TiO2 -> With the experimental set-up used (Zimbone, 2015), the nanoparticles produced have a spherical shape and a diameter of 34 nm sized. LA-Nps present, documented by TEM, XRD and Raman spectroscopy, a heavy defected very small anatase crystallite or a mixture of anatase and amorphous TiO2. Another characteristics is the presence of hydroxyl groups, in dried LA-Nps, that has been evidenced in FTIR spectroscopy. All the results here resumed you can find in the research articles (Zimbone et al. 2015; 2016a; 2016b).
  6. In Figures 6-8, how to judge the self-cleaning activity of two kinds of TiO2nanoparticles by the Δh values? And how do you know that the adhesive has an effect on the self-cleaning performance? -> According to Oleari et al., 2016, the Δh value allows quantifying information at the same time on a* (red-green colors) and b* (yellow-blue colors) and it represents the best quantity to evaluate the change in hue. This is because the adding of the blue dye induced a change hue that can be monitored in the photocatalytic test by this quantity.

The Figures 6-8 is aimed at comparing the two types of nanoparticles to examine if the laser ablated, with their specific characteristics, can be used on painting. The SA Nps are composed by anatase crystalline titanium dioxide and it is expected that they are more efficient in reaching the final h value with respect to the pure painting. This is observed for casein but not, in the same terms, for linseed oil and for egg tempera. This induced to think that the binder plays a role in the photocatalytic test. We assumed for the nanoparticles distributions on the different type of surface paintings but it is necessary to further investigations. We thank the reviewer for the useful comment. We improved the discussion of the figures 6-8 in the paper.

  1. There is no logic in the conclusion, and it’s difficult to see the key points that you want to express and a systematic summary of this research. Please clarify what this research ultimately intends to express, and systematically summarize the characterization results that can explain it - >We revised the conclusion

Reviewer 2 Report

The manuscript “Evaluation of the photocatalytic activity of water based TiO2 nanoparticles dispersions applied on historical painting surfaces” concerns the study of the photocatalytic properties of different water-based nanoparticles principally by using a colorimetric approach. The study appears interesting and useful in the field of cultural heritage. However, in my opinion, the paper is suitable for publication in Heritage after major revision. In fact, some important questions should be satisfied to improve the study (see comment below): Materials and methods  • The authors applied the same procedure to obtain Laser Ablated nanoparticles of ref [29-32]. They affirm that synthesis results are nanoparticles with 30 nm as Feret diameters measured by SEM and Dynamic Light Scattering (not reported in the manuscript), having disordered structure (amorphous phase). Structural characterization of these samples should be reported to confirm the synthesis process.  Also for crystalline nanoparticles from Sigma Aldrich, the phase is not declared and should be determined. This point of view is important because the photocatalytic properties of TiO2 nanoparticles depend on their structure: the anatase phase is generally more efficient than other phases. Amorphous TiO2 seems to shift its efficiency towards the visible region. Since this article aims to study the differences between amorphous and crystalline TiO2, in my opinion, this topic should be treated in the introduction section.   • Are the authors able to estimate the thickness of the TiO2 layer? Are they sure that the deposition is comparable?     3.2 Photocatalytic activity • Figures 6-8 and text: UV exposure induces a Δh variation also in samples with MB (e.g. in casein + MB Δh is about 30). How this question is considered in the study? What is attributable to SA or LA and what to MB? • Rows 282-284: from experimental data of figures 6-8 it does not seem that LA photocatalytic activities would be better than SA. It could be possible in terms of speed and “saturation time”. But if we observe the parameter Δh, SA presents a higher value in each sample. In addition, the SA sample, instead of LA, reaches the proximity of the final h value in casein with respect to the pure painting (without MB and TiO2). In linseed oil, they present the same final value, while for egg sample the trend is probably inverted. The authors should analyze better this trend in the text. Line 340-342 contradicts lines 282-284.   3.3 Painting surface degradation Eq. 3 is an empiric model or does exist a reference that justifies this behavior? Figure 9b (and values in the table) does not report the trend of the exponential curve for SA in egg tempera. There are some hypotheses about this? Other minor comments: Figure 5 too small, it is not clear the trend of a single curve because the colors are very similar. Figures 7 and 8 are not clear (low resolution); Line 315: wrong word wrap

Author Response

Reviewer 2

The manuscript “Evaluation of the photocatalytic activity of water based TiO2 nanoparticles dispersions applied on historical painting surfaces” concerns the study of the photocatalytic properties of different water-based nanoparticles principally by using a colorimetric approach. The study appears interesting and useful in the field of cultural heritage. However, in my opinion, the paper is suitable for publication in Heritage after major revision. In fact, some important questions should be satisfied to improve the study (see comment below).

We thank the reviewer for the comments. They were useful to improve the paper. We replied below, point by point, to the questions.

Materials and methods.

  • The authors applied the same procedure to obtain Laser Ablated nanoparticles of ref [29-32]. They affirm that synthesis results are nanoparticles with 30 nm as Feret diameters measured by SEM and Dynamic Light Scattering (not reported in the manuscript), having disordered structure (amorphous phase). Structural characterization of these samples should be reported to confirm the synthesis process.- > With the experimental set-up used (Zimbone, 2015), the nanoparticles produced have a spherical shape and a diameter of 34 nm sized. LA-Nps present, documented by TEM, XRD and Raman spectroscopy, a heavy defected very small anatase crystallite or a mixture of anatase and amorphous TiO2. Another characteristics is the presence of hydroxyl groups, in dried LA-Nps, that has been evidenced in FTIR spectroscopy. All the results here resumed you can find in the research articles (Zimbone et al. 2015; 2016a; 2016b).
  • Also for crystalline nanoparticles from Sigma Aldrich, the phase is not declared and should be determined. This point of view is important because the photocatalytic properties of TiO2 nanoparticles depend on their structure: the anatase phase is generally more efficient than other phases. ->The crystalline nanoparticles from Sigma Aldrich was characterized in previous articles of the same authors. See reference Zimbone et al. 2015. They are composed by anatase phase.
  • Amorphous TiO2 seems to shift its efficiency towards the visible region. Since this article aims to study the differences between amorphous and crystalline TiO2, in my opinion, this topic should be treated in the introduction section. -> We added the topic in the introduction.
  • Are the authors able to estimate the thickness of the TiO2 layer? Are they sure that the deposition is comparable? -> RBS measurements and SEM in cross section analyses have been scheduled.

3.2 Photocatalytic activity

  • Figures 6-8 and text: UV exposure induces a Δh variation also in samples with MB (e.g. in casein + MB Δh is about 30). How this question is considered in the study? What is attributable to SA or LA and what to MB?

According to literature, the MB dye is sensitive to UV exposure. This is not ignored in this study and for this reason, we reported both the trends with only MB and with MB+nanoparticles. The graphs showed that, the degradation rate with nanoparticles is enhanced. In fact, if Δh is about equal to30 for casein + MB at the end of the test, in the case of casein + MB + TiO2 the Δh is about 60 that is the double (about 60). We thak the reviewer for the useful comment, the data were better discussed in the paper.

  • Rows 282-284: from experimental data of figures 6-8 it does not seem that LA photocatalytic activities would be better than SA. It could be possible in terms of speed and “saturation time”. But if we observe the parameter Δh, SA presents a higher value in each sample. In addition, the SA sample, instead of LA, reaches the proximity of the final h value in casein with respect to the pure painting (without MB and TiO2). In linseed oil, they present the same final value, while for egg sample the trend is probably inverted. The authors should analyze better this trend in the text.

We did not affirm that the LA Nps are better that SA ones. The aim is to compare the two types of nanoparticles to examine if the laser ablated, with their characteristics, can be used on painting. The SA Nps are composed by anatase crystalline titanium dioxide and it is expected that they are more efficient in reaching the final h value with respect to the pure painting. This is observed for casein, as the reviewer said, but not for linseed oil and egg tempera. This induced to think that the binder plays a role in the photocatalytic test. We assumed for the nanoparticles distributions on the different type of surface paintings but it is necessary to further investigations. We thank the reviewer for the useful comment. We improved the discussion of the figures 6-8 in the paper.

Line 340-342 contradicts lines 282-284. -> Revised

3.3 Painting surface degradation Eq. 3 is an empiric model or does exist a reference that justifies this behavior? -> The Equation 3 is a empiric model but it can find justification in literature. It is known, in fact, that the fading of dye or pigment often obeys to exponential decay behaviour (Feller RF, Accelerated Aging. Photochemical and Thermal Aspects, 1994, J Paul Getty Museum Pubs). This information and the reference have been added in the article.

Figure 9b (and values in the table) does not report the trend of the exponential curve for SA in egg tempera. There are some hypotheses about this? ->Egg tempera is a binder composed by whole egg and Arabic gum, as reported in the compositional sheet of the binder provided by the company. This could led to a complex degradation rate that takes into account also the contribution of more components. The empiric model, found for C* color coordinate, in this case, cannot fit the behaviour of a more composite system than casein and linseed oil. This comment is rewritten and it is added in the text.

Other minor comments:

Figure 5 too small, it is not clear the trend of a single curve because the colors are very similar. -> Revised

Figures 7 and 8 are not clear (low resolution) -> Revised

Line 315: wrong word wrap. -> Revised

Reviewer 3 Report

The title of the manuscript is in line with the paper`s contents.

The abstract is fully proper, including the justification of a research, experimental characterization, used techniques, obtained results and concluding remark.

The keywords are proper.

As concerns the introduction, it is of an appropriate length and counts sufficient number of references. The subject and aim of the research are well described.

In MATERIALS AND METHODS section, I find some small errors:

Line 106: reference: it should be in plural form, references.

Line 115: please add the names, producer and country for both DLS and SEM.

Line 128: Bianco di San Giovanni” pigment; please add the delivering company or producer and country.

Besides, I have no other comments on this section.

In RESULTS AND DISCUSSION section, I have following remarks:

Table 2: please change in numbers the commas to points, e.g., 0.50 instead of 0,50

CONCLUSIONS section seems too large and to a great extent is a summary of results. Make it more comprehensive, and more limited to the results and their sources, and not to the experimental details.  

Author Response

The title of the manuscript is in line with the paper`s contents.The abstract is fully proper, including the justification of a research, experimental characterization, used techniques, obtained results and concluding remark. The keywords are proper.As concerns the introduction, it is of an appropriate length and counts sufficient number of references. The subject and aim of the research are well described.

We thank the reviewer for the comments. They were useful to improve the paper. We replied below, point by point, to the questions.

In MATERIALS AND METHODS section, I find some small errors:

Line 106: reference: it should be in plural form, references. à Revised

Line 115: please add the names, producer and country for both DLS and SEM. à Revised. SEM name is added. The DLS measurements are performed with a homemade apparatus described elsewhere (Zimbone et al., 2015).

Line 128: Bianco di San Giovanni” pigment; please add the delivering company or producer and country. à Added

Besides, I have no other comments on this section.

In RESULTS AND DISCUSSION section, I have following remarks:

Table 2: please change in numbers the commas to points, e.g., 0.50 instead of 0,50à Revised

CONCLUSIONS section seems too large and to a great extent is a summary of results. Make it more comprehensive, and more limited to the results and their sources, and not to the experimental details. -> We revised the conclusion

Round 2

Reviewer 1 Report

The author answered all the questions and revised the manuscript, can be accepted now.

Reviewer 2 Report

All the questions proposed in my previous report have been satisfied by the authors. The paper can be accepted in the present form.